# LncRNAs Are Differentially Expressed between Wildtype and Cell Line Strains of African Trypanosomes

**DOI:** 10.3390/ncrna8010007

**Published:** 2022-01-12

**Authors:** Hyung Chul Kim, Emmitt R. Jolly

**Affiliations:** 1Department of Biology, Case Western Reserve University, Cleveland, OH 44106, USA; hxk298@case.edu; 2Center for Global Health and Disease, Case Western Reserve University, Cleveland, OH 44106, USA; 3Center for RNA Science and Therapeutics, Case Western Reserve University, Cleveland, OH 44106, USA

**Keywords:** trypanosoma, lncRNA, non-coding RNA, parasite, differential expression, splicing

## Abstract

*Trypanosoma brucei* is a parasitic protist that causes African sleeping sickness. The establishment of *T. brucei* cell lines has provided a significant advantage for the majority of *T. brucei* research. However, these cell lines were isolated and maintained in culture for decades, occasionally accumulating changes in gene expression. Since trypanosome strains have been maintained in culture for decades, it is possible that difference may have accumulated in fast-evolving non-coding RNAs between trypanosomes from the wild and those maintained extensively in cultures. To address this, we compared the lncRNA expression profile of trypanosomes maintained as cultured cell lines (CL) to those extracted from human patients, wildtype (WT). We identified lncRNAs from CL and WT from available transcriptomic data and demonstrate that CL and WT have unique sets of lncRNAs expressed. We further demonstrate that the unique and shared lncRNAs are differentially expressed between CL and WT parasites, and that these lncRNAs are more evenly up-regulated and down-regulated than protein-coding genes. We validated the expression of these lncRNAs using qPCR. Taken together, this study demonstrates that lncRNAs are differentially expressed between cell lines and wildtype *T. brucei* and provides evidence for potential evolution of lncRNAs, specifically in *T. brucei* maintained in culture.

## 1. Introduction

*Trypanosoma brucei* is the causative agent for African trypanosomiasis, a neglected tropical disease that leads to sleep disturbance and is almost always fatal if untreated [1]. The disease is transmitted by the bite of the tsetse fly, during which the protozoan *T. brucei* invades the mammalian host and enters the bloodstream. Mammalian host specificity depends on the subspecies of the parasite: *T. brucei rhodesiense* (Tbr) and *T. brucei gambiense* (Tbg) infect human hosts, while *T. brucei brucei* (Tbb) primarily infects cattle. Tbb is unable to survive in humans due to its susceptibility to trypanolytic factors (TLF) present in human sera [2]. Tbr overcomes this susceptibility by expressing serum resistance-associated (SRA) gene [3,4] while Tbg uses multiple independent components to nullify TLF [5,6,7,8,9]. Despite these differences, whole genome sequencing demonstrates that the human-infecting Tbr and cattle-infecting Tbb are highly similar [10].

In addition to causing debilitating disease, *T. brucei* has a distinctly unique evolutionary lineage and is an interesting research organism [11]. The establishment of a cell line for in vitro study has been crucial for trypanosome research [12,13]. However, the most commonly used cell lines for *T. brucei* studies are sourced from isolates collected decades ago [14]. In fact, the first trypanosome cell line to have its genome sequenced, Tbb TREU927, was isolated over 50 years before it was sequenced [14,15]. Furthermore, while cell lines allow for an in-depth study at the in vitro level, they may behave very differently from in vivo conditions. Cell lines in humans have been shown to exhibit a significant difference from in vivo samples [16,17], and in similar fashion, different *T. brucei* cell lines vary in their expression of RNA editing pathways [18]. Additionally, transcriptomic analysis between Tbr from human patients and cell-cultured Tbb shows that cultured trypanosomes have an expression profile that suggests faster proliferation when compared to human blood samples [19]. While the previous transcriptional analysis established differences in protein-coding genes, it did not also account for non-coding RNA.

Long non-coding RNA (lncRNA) is a class of non-coding RNA that is over 200 nucleotides in length with little to no protein-coding potential [20]. LncRNAs have diverse functions in gene regulation and interact with DNA, RNA, proteins, and chromatin [21,22,23,24,25,26,27]. In *T. brucei*, one lncRNA has been shown to have a functional role in parasite differentiation during the bloodstream stage [28]. LncRNAs have been reported to have faster evolutionary rates than protein-coding genes [29]. While lncRNA sequences and expression patterns evolve more rapidly than protein-coding genes, their tissue-specificity is usually conserved [30]. Although lncRNA sequences are not as conserved as protein-coding regions, they are more conserved than introns and random intergenic regions [31,32]. Thus, lncRNAs offer a unique opportunity to examine how decades of culturing has impacted *T. brucei*. In our study, we pooled *T. brucei* RNA-seq datasets from both cell line (CL) and wildtype (WT) sources and used two lncRNA identification tools to identify a high-confidence set of lncRNAs. We then show that the CL and WT strains each has a unique set of lncRNAs and also share common lncRNAs that have different expression patterns compared to protein-coding genes. We further verify the expression of newly identified lncRNA expression using quantitative RT-PCR.

## 2. Results

### 2.1. Novel lncRNAs Were Identified in Both CL and WT T. brucei

A vast majority of the available *T. brucei* cell lines were isolated decades ago [14], and have been extensively used for *T. brucei* research, including transcriptomic analysis. While RNA-seq has been performed several times in CL *T. brucei* [19,33,34,35,36], WT *T. brucei* has rarely been investigated using transcriptomic analysis. Despite recent interest in the roles of lncRNAs, a thorough comparison of the differences in these RNAs has not been performed. To compare lncRNAs from CL and WT *T. brucei*, we used thirty-three RNA-seq datasets from three published transcriptomic studies for lncRNA identification [19,33,34]. An RNA-seq dataset was considered CL if the study used an established *T. brucei* cell line in culture or in mice. On the other hand, an RNA-seq dataset was considered WT if the source parasites were recently collected from human blood or cerebrospinal fluid (CSF). For the CL studies, any datasets from the procyclic form of *T. brucei* was discarded, as no WT dataset is available for this life stage. For the WT datasets, any dataset that had lower than 1% alignment rate to the *T. brucei* TREU927 genome was discarded. This criterion was chosen as the WT samples collected directly from patients were contaminated with host tissue, and the published work showed that 1–10% of the reads were trypanosome-specific [19]. Overall, nine runs of CL and 24 WT datasets were used for this process (Appendix A). The principal-component analysis showed that the CL datasets and WT datasets form two major clusters with subclusters in each group due to heterogeneous nature of the data (Appendix A).

Next, we designed a computational pipeline to identify potential lncRNAs from these datasets at high-confidence level by using two different lncRNA identification tools. After quality checking and trimming the adapters, the RNA-seq datasets were aligned to the *T. brucei* TREU927 genome and de novo assembled into transcripts. These assemblies were merged, and rRNAs and transcripts matching spliced-leader RNA (SL-RNA) were removed from the assembly. The resulting assembly then underwent the lncRNA identification process using two different tools (Figure 1). One of the tools, FEELnc, chosen is a high-performing program optimized for non-model organisms [37], and the second, CPC2, is a species-neutral lncRNA identification tool that has been used in non-model organisms [38,39,40,41]. For FEELnc, to increase the sensitivity and specificity, we used the shuffle mode of the coding potential calculation with a relaxed ORF definition, multi k-mer frequencies, and increased the mRNA and lncRNA specificity thresholds to 0.99 [37]. Additionally, the mono-exonic lncRNAs were included in this process, as *T. brucei* largely does not undergo cis-splicing, and previous work in various organisms have identified mono-exonic lncRNAs [42,43,44,45]. The results from FEELnc and CPC2 were then compared to select only the overlapping lncRNAs that were identified by both tools, resulting in a reduction of potentially identified lncRNAs. These high-confidence potential lncRNAs were then quantified using Salmon in mapping-based mode for further differential expression analysis [46].

LncRNAs identified through this process included novel transcripts and transcripts that have already been annotated in the *T. brucei* TREU927 genome as mRNAs, pseudogenic transcripts, and ncRNAs. Despite being classified as mRNAs, most of these transcripts are annotated as hypothetical proteins or putative genes that have not been experimentally verified. Since the lncRNA identification tools were used with a higher sensitivity and specificity threshold, we decided to retain these transcripts except for genes annotated with more defined protein domains in their gene names. However, we retained any transcripts that have been annotated as putative variant surface glycoprotein (VSG) or expression site-associated gene (ESAG) as most of these transcripts are annotated as pseudogenes, degenerates, or frameshifted. With these criteria, we removed 43 transcripts previously classified as mRNAs from the lncRNA list. In the past, pseudogenes have been regarded as “junk genes” that were considered defunct pieces of protein-coding genes. Nevertheless, some pseudogenes have now been shown to produce lncRNAs that can have further regulatory functions [47,48]. With the possibility of pseudogene-derived lncRNAs, we retained the transcripts that have been previously classified as pseudogenic transcripts. The transcripts that have been previously classified as ncRNA were also retained in this analysis, as their lengths were sufficient to classify them as lncRNAs. Using this approach, we identified 978 potential lncRNAs composing of 535 novel transcripts, 152 transcripts annotated as mRNAs, 246 annotated pseudogenic transcripts, and 45 annotated ncRNAs. With these potential lncRNAs identified, we then investigated whether they were differentially expressed between CL and WT *T. brucei*.

#### 2.1.1. CL and WT *T. brucei* lncRNA Expression Patterns Are Distinct

In addition to CL *T. brucei* being isolated decades ago, most CL *T. brucei* were collected from ungulates [14]. We speculated that some of the newly identified potential lncRNAs might be expressed exclusively in CL or in WT. To test this, the transcripts were quantified and normalized for gene expression comparison, and only the transcripts with stringent, average normalized count over 10 were included for this analysis. For comparison, we also performed the same analysis for protein-coding genes. Among the 978 potential lncRNAs, 112 (11.4%) transcripts were expressed only in the CL, 82 (8.4%) transcripts were expressed only in the WT, and 457 (46.7%) transcripts were expressed in both CL and WT (Figure 2A). The remaining transcripts were expressed, but at low level in both strains. Compared to the lncRNAs, protein-coding genes showed more conservation. Among 9729 protein-coding transcripts, 185 (1.9%) transcripts were expressed only in the CL, 394 (4.0%) transcripts were expressed only in the WT, and 8426 (86.6%) transcripts were expressed in both CL and WT (Figure 2B). However, if we used a normalized count over 1 for inclusion, we identified 61 (6.2%) lncRNAs expressed only in the CL, 111 (11.3%) lncRNAs expressed only in the WT, and 646 (66.1%) lncRNAs expressed in both CL and WT (Appendix A). For protein-coding transcripts with normalized count over 1, 65 (0.7%) transcripts were expressed only in the CL, 356 (3.7%) transcripts were expressed only in the WT, and 8886 (91.3%) transcripts were expressed in both CL and WT (Appendix A). This analysis suggests that there has been greater divergence of CL and WT strain lncRNAs as compared to the protein-coding genes. The top 15 expressed CL, WT, and shared transcripts are highlighted in Appendix A.

While these unique sets of lncRNAs can have potentially important roles in either strain, the differential expression of the common sets of lncRNAs can also have a significant role in gene regulation. We used differential expression analysis to compare the log2 fold change (log2FC) of the WT to the CL. Among the 978 identified lncRNAs, 139 (14.2%) transcripts were up-regulated in WT while 133 (13.6%) transcripts were down-regulated in WT when compared to the CL (*p* < 0.01) (Figure 3A). We then compared this result to the differential expression of protein-coding transcripts. Among the 9729 protein-coding transcripts in *T. brucei* TREU927 genome, 1155 (11.9%) were up-regulated and 2855 (20.3%) were down-regulated in the WT when compared to the CL (*p* < 0.01) (Figure 3B). The broadly even up-regulated and down-regulated pattern of lncRNA compared to the largely down-regulated pattern of protein-coding transcripts in WT suggests that the lncRNAs have a more evenly up-regulated and down-regulated expression pattern in both strains when compared to protein-coding genes.

#### 2.1.2. RT-qPCR Assay Confirms the In Silico lncRNA Expression in *T. brucei*

To validate the lncRNA identification pipeline, we performed qPCR analysis on several identified lncRNAs. However, after the data were assembled using Stringtie, we were quite surprised. The data predicted that a significant number of lncRNA were potentially spliced. We questioned the validity of the output, and consequently performed a de novo assembly of Tb protein coding genes where only two spliced genes are already known. Using the same analysis, our data were consistent and supported the currently established two spliced protein coding genes. We selected five lncRNAs with unique sequences for RT-qPCR analysis. Of the five lncRNAs, three were mono-exonic lncRNAs and we identified two as potentially multi-exonic lncRNAs. Tb927.11.10190 (telomerase reverse transcriptase) was used as the reference gene [49]. Expression was detected for all five transcripts, with mono-exonic lncRNAs expression being higher than the reference gene and the multi-exonic lncRNAs expression at similar or lower level than the reference gene (Figure 4). In addition, the melt curves showed one peak for each qPCR sample (Appendix A). All five expressed lncRNAs showed very little to no genomic DNA contamination, confirming that the predicted lncRNAs are expressed in vivo.

## 3. Discussion

In this study, we pooled 33 RNA-seq datasets and used two lncRNA identification tools to identify potential lncRNAs in CL and WT *T. brucei*. We identified 978 lncRNAs comprised of 536 novel transcripts and 445 transcripts that have been annotated as mRNA, pseudogenic transcript, or ncRNA in the genome. We then provided data that both CL and WT each have a unique set of lncRNAs that are expressed exclusively in each strain, and that lncRNAs are differentially expressed between CL and WT. Further, we compared lncRNA differential expression to protein-coding genes to show that lncRNAs are proportionally more evenly up-regulated and down-regulated while protein-coding genes are largely down-regulated in WT when compared to CL. Finally, we verified the expression of these lncRNAs using RT-qPCR. Our findings support that cell lines have a distinct class of differentially genes when compared to non-cultured samples [16,17]. The differential expression analysis also bolsters the idea that lncRNAs are an ideal standard to assess the effect of in vitro culturing compared to in vivo samples.

A strange finding in the identified lncRNAs is that our data suggest that many of these transcripts are spliced. Trypanosomatids are known to have very few verified cis-spliced transcripts [50,51,52]. Trypanosome transcripts primarily undergo trans-splicing of SL-RNA to the 5′ cap of the polycistronic precursor to mature [50,53]. The known minimum eukaryotic intron length is 30 bp [54]. Our analysis found that 403 out of the 978 potential lncRNAs are multi-exonic by this definition, of which 400 are novel transcripts and three are pseudogenic transcripts. We tested two of the potentially multi-exonic lncRNAs by RT-qPCR that targeted the exon junctions (Figure 4). These data seemed to verify, with melt curves that showed a single peak, that trypanosome may indeed have additional cis-spliced genes among their lncRNA populations (Appendix A). Despite the dearth of verified cis-spliced transcripts, the *T. brucei* TREU927 reference genome actually annotates many multi-exonic transcripts. Out of 9881 transcripts labeled as protein-coding genes, four transcripts have multiple exons. Furthermore, of the 1383 transcripts labeled as pseudogenes, 26 transcripts are annotated with multiple exons. The presence of a predicted higher number of multi-exonic transcripts in the pseudogenic transcripts, which can potentially lead to pseudogene-derived lncRNAs [47,48], suggests that multi-exonic lncRNAs may exist in *T. brucei*. Furthermore, the RT-qPCR for multi-exonic lncRNAs confirmed their expression despite the fact that the primers were designed to target exon junction that are thousands of base pairs apart. Therefore, the data suggest the existence of multi-exonic lncRNAs in *T. brucei.* Further investigation is in underway.

Previous transcriptomic analysis in cultured and bloodstream/CSF suggested that cultured trypanosomes potentially have faster division times than human-sourced trypanosomes due to higher levels of RNA-binding proteins, cytoskeletal proteins, mitochondrial proteins, translation initiation factors, and proteasome subunits [19]. Our analysis of protein-coding genes concurs that WT strains have a lower expression of the genes listed above, along with a large proportion of protein-coding genes. However, the differential expression pattern in protein-coding genes differs from that of lncRNAs. While protein-coding genes were largely down-regulated in WT, lncRNAs were proportionally up-regulated and down-regulated (Figure 3). Moreover, we found that both CL and WT strains each express a unique set of lncRNAs: 112 (11.4%) transcripts expressed only in the CL, 82 (8.4%) transcripts expressed only in the WT. Such discrepancy in the expression level and the unique transcripts support the idea that lncRNAs have faster evolutionary rate than protein-coding genes [29].

The existence of unique lncRNAs does not necessarily require that they have unique functions in CL or WT. LncRNAs are more conserved in terms of tissue-specificity [30], and both CL and WT are bloodstream forms of *T. brucei*. Furthermore, lncRNA conservation has multiple levels: sequence, structure, function, and syntenic context of neighboring genes [55]. Therefore, some of these unique lncRNAs in CL and WT may be conserved at the structural level, or they could be functional counterparts with different sequences and structures. For example, the human lncRNA JPX and the mouse lncRNA Jpx have different sequences and secondary structures, but both bind the same protein partner during X-chromosome inactivation [56]. Likewise, some of the potentially unique lncRNAs identified may have evolved new sequences or completely degenerated through numerous cell culture passages, leaving behind structural remains emblematic of functional cousins. Only one lncRNA has been functionally analyzed in *T. brucei*. The lncRNA, *grumpy*, controls differentiation from proliferating slender form to non-proliferating stumpy form in bloodstream *T. brucei* [28], signifying that lncRNAs are crucial in parasite biology. An important set of “mRNAs” that were retained as lncRNAs were annotated in the genome as degenerate and frame-shifted forms of variant surface glycoproteins (VSGs), which are essential for evasion of the host immune system [57]. In terms of lncRNA conservation, these transcripts could function in regulating VSG-switching to evade the host immune system, and have potential as therapeutic targets.

## 4. Materials and Methods

### 4.1. Genomic and Transcriptomic Data Analysis

The *T. brucei* TREU927 genome sequence and annotation were downloaded from TriTrypDB [58]. The most recent version, release 54, of the genome was used for this analysis.

Thirty-three RNA-seq datasets from three studies were used for this study: (1) four runs from cultured bloodstream form *T. brucei* RNA-seq datasets (PRJEB19907) [33], (2) five runs of CL *T. brucei* grown in mice and collected from blood and fat tissue (PRJEB11801) [34], and (3) twenty-four runs selected from WT *T. brucei* collected from human patients (PRJEB18524, PRJEB18523, and PRJEB23278) [19]. For the WT datasets, sixteen runs were directly collected from humans, either from blood or CSF, while eight runs were collected from human patients, injected into rats, and collected back again. The four cultured blood stream forms are *T. brucei brucei* EATRO 1125, clone AnTat1.1 strain while the five CL parasites were *T. brucei brucei* EATRO 1125 Antat1.1^E^ 90–13 strain. The WT parasites were *T. brucei rhodesiense* collected from patients from Kaberamaido district of North-Eastern Uganda. FastQC was used on all datasets to check for the dataset quality [59] (https://www.bioinformatics.babraham.ac.uk/projects/fastqc/, accessed on 8 January 2020), and Trimmomatics was used to trim any adapters present in the datasets [60]. Next, the dataset was aligned to the *T. brucei* TREU927 genome using STAR [61] (https://github.com/alexdobin/STAR, accessed on 2 October 2020) and de novo assembled into transcripts using Stringtie [62]. Any RNA-seq runs with less than 1% alignment rate to the genome were discarded, resulting in the number of datasets mentioned above. The individual assemblies were then merged using Stringtie with the genome as the reference to generate a single GTF file for downstream analysis.

### 4.2. Identification of Long Noncoding RNAs

The merged assembly was processed through riboPicker to identify and remove any rRNAs [63] (http://ribopicker.sourceforge.net/, accessed on 5 June 2020). The following databases were used for riboPicker: SILVA rRNA database for large subunit (version 132) and small subunit (version 138) rRNA sequences [64] (https://www.arb-silva.de/, accessed on 5 June 2020) and Rfam database for 5S and 5.8S subunit rRNA sequences (release 14.2) [65] (https://rfam.org/, accessed on 5 June 2020). Since the SILVA small subunit data file was too large to index with riboPicker, it was broken into six smaller sized files for indexing.

After the rRNA filtering, the assembly was processed with two different lncRNA identification tools: FEELnc and CPC2. The FEELnc coding potential calculation module was performed on shuffle mode with relaxed ORF definition and multi k-mer frequencies to increase sensitivity of the module [37] (https://github.com/tderrien/FEELnc, accessed on 16 March 2020). To decrease the false positive rate due to the optimized sensitivity, the specificity thresholds for mRNA and lncRNA were each increased to 0.99. CPC2 was run on the default setting [38] (http://cpc2.gao-lab.org/, accessed on 10 September 2020). The transcript ID was used to find the lncRNA transcripts that were shared by the results from both lncRNA identification tools.

### 4.3. Statistical Analyses

The transcripts were quantified using the mapping-mode of Salmon [46] with the identified lncRNAs as reference transcripts. The protein-coding gene quantification was done with all annotated mRNA from the *T. brucei* TREU927 genome with the 152 transcripts identified as lncRNAs removed from the reference. Statistical analyses were performed with R environment (v. 4.0.5 [66]) with DESeq2 [67], readr [68], and tximport [69] libraries loaded. The log2 fold change shrinkage was performed with the apeglm [70] package. The *p* values used for all analyses were adjusted for multiple testing.

### 4.4. Animals and Parasites

Bloodstream form *T. brucei brucei* Lister 427 were cultured in T75 vented cap culture flasks (NEST, Wuxi, China) in complete HMI-9 media (Axenia Biologix, Dixon, CA, USA) containing 10% FBS (VWR, Radnor, PA, USA), 10% Serum Plus medium supplement (MilliporeSigma, Burlington, MA, USA), and 1X penicillin-streptomycin (Thermo Fisher, Waltham, MA, USA). The trypanosomes were maintained in a humidified incubator at 37 °C and 5% CO_2_, and they were sub-cultured every 48 h. After checking for parasite density for log growth phase, the trypanosomes were collected, pelleted, and flash frozen.

### 4.5. RNA Extraction and cDNA Generation

RNA was extracted from frozen pellets of *T. brucei* using the Direct-zol RNA Miniprep Kit (Zymo Research, Irvine, CA, USA) with a minor modification. To decrease the genomic DNA contamination, on-column DNase I treatment was performed twice. RNA concentration and quality were assessed on a Nanodrop 8000 Spectrophotometer (Thermo Scientific, Waltham, MA, USA). The cDNA generation was performed with Superscript II Reverse Transcriptase kit (Thermo Scientific, Waltham, MA, USA) with oligo dT primer following the manufacturer’s protocol.

### 4.6. RT-qPCR Assays

The qRT-PCR reactions were performed with Power SYBR Green RNA-to-C_T_ 1-Step Kit (Applied Biosystems, Foster City, CA, USA). using two Q-qPCR machines (QuantaBio, Beverly, MA, USA). The no-RT reactions were set up to determine DNA contamination in the RNA samples. For reference gene, Tb927.11.10190 was chosen as it was determined to be an ideal reference gene for *T. brucei* qPCR studies [49]. The experiments were performed in triplicate and ΔCT method was used to analyze the transcript expression levels relative to the reference gene. The primer sequences are included in Appendix A.

## Figures and Tables

**Figure 1 ncrna-08-00007-f001:**
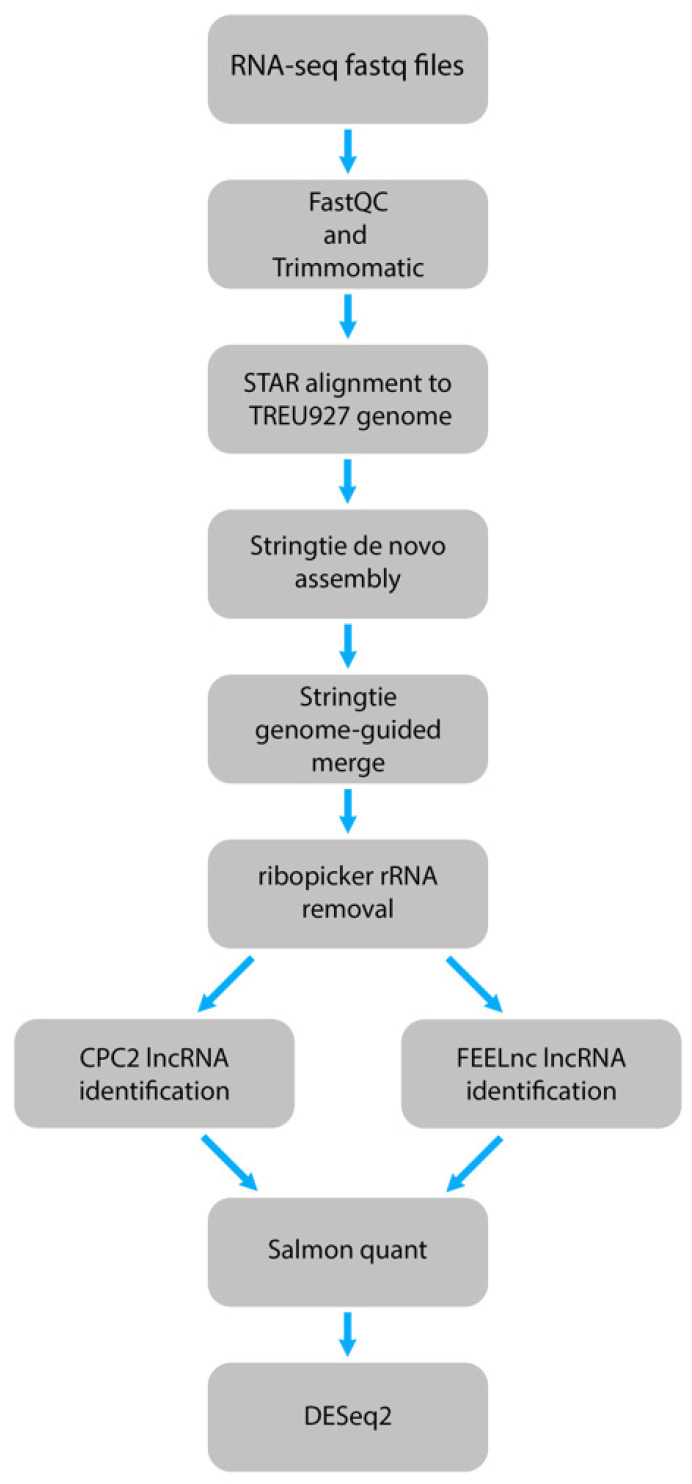
Schematic of the computational pipeline designed to identify lncRNAs from pooled *T. brucei* RNA-seq datasets using two lncRNA identification tools.

**Figure 2 ncrna-08-00007-f002:**
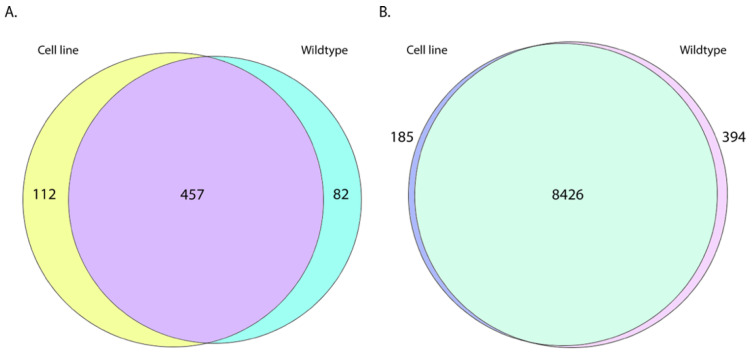
Venn diagram of (**A**) identified *T. brucei* lncRNA and (**B**) protein-coding gene expression pattern in CL and WT strains. Only the transcripts with average >10 DESeq2 normalized count were included for this analysis.

**Figure 3 ncrna-08-00007-f003:**
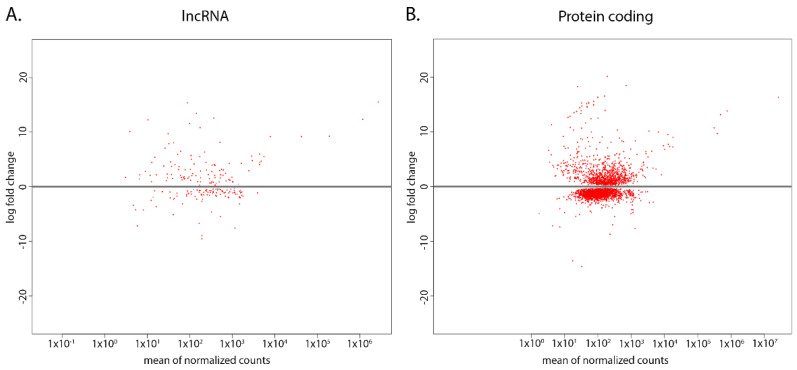
LncRNAs are differentially expressed between CL and WT *T. brucei*. Red dots indicate transcripts that are significantly differentially expressed (*p* < 0.01), and the log fold change of WT transcripts are compared to that of CL strain. (**A**) MA-plot of the identified lncRNAs in WT compared to CL. (**B**) MA-plot of protein-coding transcripts in WT compared to CL.

**Figure 4 ncrna-08-00007-f004:**
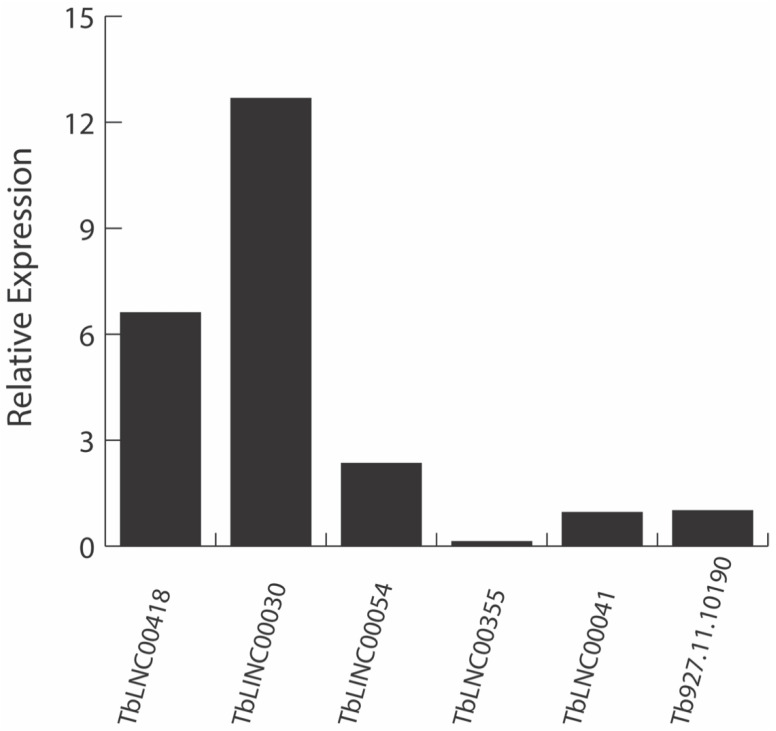
RT-qPCR confirms the expression of identified lncRNAs. Five lncRNAs with unique sequences were selected for expression confirmation in CL *T. brucei*, and their relative expression level was calculated compared to the reference gene. TbLNC00418, TbLINC00030, and TbLINC00054 are mono-exonic transcripts. TbLNC00355, and TbLNC00041 are multi-exonic transcripts. Tb927.11.10190 is reference gene.

## Data Availability

All data analyzed in this study can be found in NCBI’s Sequence Read Archive under Accession numbers PRJEB19907, PRJEB11801, PRJEB18523, PRJEB18524, and PRJEB23278, and further included in Appendix A.

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
