# Peer review of "LncRNAs Are Differentially Expressed between Wildtype and Cell Line Strains of African Trypanosomes"

_ncrna, 2022, doi:10.3390/ncrna8010007_

Round 1

Reviewer 1 Report

The manuscript entitle "LncRNAs are differentially expressed between wildtype and cell line strains of African trypanosomes" (ID: ncrna-1498849) was reviewed carefully. Some comments raised as following:

  • Scientific name of parasite species must be written in italic, especially in the abstract section.
  • "in vitro" and "in vivo" are written in italic. Check the manuscript.
  • Line 55: add this ref to (21-26) as well: (PMID: 31208095).
  • The link for FEELnc, CPC2, FastQC, STAR, SILVA, and riboPicker tools must be mentioned in the parenthesis and immediately the relevant citation.

Reviewer 2 Report

The manuscript presented by Kim and Jolly focus on the re-analysis of RNA-seq data derived from patient isolates (WT) and stablished laboratory strain of T. brucei (CL). They apply LncRNAs bioinformatics detection through two different algorithms (CPC2 and FEELnc) to test the hypothesis, implying that lncRNA expression may be different between strains maintained many years in the lab versus parasites extracted from infected hosts. This reviewer finds the idea and the results interesting and well carried out, inspiring the following questions or possible improvement to the way the data is presented.

  1. First, it is understandable that the authors chose to produce the list of LncRNA from pooled data of WT or CL. It is not that obvious, as strategy, to pool the two types (WT and CL) into two single categories to perform gene expression analysis. Is it possibly to obtain more differentially expressed LncRNAs studying individual pairs of datasets? i.e., are all the WT datasets comparable? Are the authors losing information due to averaging? Perhaps a PCA analysis including all samples should be provided to identify distances among different datasets. Supplementary data is provided but rather scarce. More metadata associated to each data set is necessary (type of T. brucei group cell lines (brucei, gambiense, rhodesiense?) and if possible, for patients, patient demographics, disease stage, culture conditions of the cell lines, purification methods); anything that may point to difference in the methods, or the parasites isolated in each case and has not relation to the CL-WT status. This can be of use to understand the number of LncRNA obtained and differentially expressed. likely will be better to name these LncRNA not “unique”, but “Highly expressed” in WT or CL derived conditions.
  2. Figure 2 its very interesting and it’s the base to present unique LncRNA in WT or CL. Unfortunately, it’s possible that the uniqueness is derived from the cut off of 10 counts per LncRNA used to build the list to be crossed by the Venn diagram. It would be interesting to show an additional panel with different cutoff (lower: 1 count and higher: 50 counts). The later will not strike out the main concept of the paper, just widen the view for the non-expert viewer.
  3. Supplementary material must include information at least of the top 10-15 LncRNAs highly expressed (here termed “unique” in WT and CL) and a table of the top 10 differentially expressed LncRNA, that can be evaluated and interrogated by peers. If they were new, not previously annotated, sequence information will be essential.

4             The claim made in the discussion about an astonishing amount of splicing (403 out of 978 lncRNAs) in a kinetoplastid, that are known to only processes few genes by cis-splicing needs at least a figure in result section because it is quite unexpected.

Overall, the organization of the result presentation and the fluency of the text are appropriate, and the subject is interesting and original. Nevertheless, the findings need to be presented in a more thorough and transparent (not closed) fashion to allow the Reader and Reviewers to appraise the quality and significance of the data.

Reviewer 3 Report

Chul Kim and Jolly study the presence, expression, and characteristics of lncRNAs in T. brucei cell lines and wild-type parasites. This is an interesting topic that has not been addressed in the literature. So, I consider the results worth exploring and published. However, I have some concerns that I hope the authors can address before publication.

Major comments:

1 - Did do you control for batch effect when comparing different experiments?  It would be useful to see a PCA analysis of all the samples to assess the possible influence of this effect on differential expression analysis.

2 - The observation about multiexonic transcripts is intriguing. How did you define multiexonic lncRNAs? Considering that this type of splicing is rare in trypanosomatids, can this be an error of Stringtie where the software is incorrectly joining two (or more) monoexonic transcripts?

3 - I fail to see the rationale behind the assertion that “lncRNAs have more dynamic expression” in the phrase “The evenly upregulated and downregulated pattern of lncRNA compared to largely downregulated pattern of protein-coding transcripts in WT suggests that the lncRNAs have more dynamic expression pattern in both strains when compared to protein-coding genes”.

4 - “To validate the lncRNA identification pipeline and the differential expression of lncRNAs, five lncRNAs with unique sequences were selected for RT-qPCR analysis. As we did not have access to WT strain, the verification was only done in RNA extracted from cultured trypanosomes”. I don’t think that you are controlling for differential expression by performing this experiment, at least not between samples. Also, it would be very useful to show the correlation between the qPCR values and the expression estimates calculated by the transcriptomic pipeline (Normalized Salmon counts).

Minor comments

5 - Figure 2 - I think that showing percentages beside the raw numbers in the figure would make the point made by the authors clearer.

Reviewer 4 Report

The manuscript by Kim and Jolly describes the bioinformatic analysis of the lncRNAs of different T. brucei strains, specifically comparing strains isolated from the field and those usually used in the lab and established decades ago. The analysis is thoroughly and interesting and could possibly provide interesting data on the possible changes accumulated after such a long time in artificial culture conditions. Despite that, several important points need to be addressed.

A critical missing point is that there is no real validation of the data. qRT-PCR was only performed on the CL cultured in supplemented HMI9 media, which is expected to be more reproducible than WT isolates. Without data for the CL in animal models and WT isolates the manuscript can only be considered a bioinformatic prediction.

Another issue is that the culturing conditions of the studied cell lines are completely different (meaning the bloodstream of humans for the WT isolates and the media or mice for the lab CL) and therefore any conclusion about gene expression is not necessarily a result of the long cultivation time but most probably adaptations to the culturing media. If both lines are not cultured under the same conditions, no real conclusion can be drawn.

The suggestion that multiexonic lncRNAs might exist in T. brucei is very interesting. Considering the authors were successful amplifying these species by qPCR they could also provide raw data about this (bands in a gel).

Minor points:

Maybe the authors could also analyze the conservation of the lncRNAs (point mutations), the length, possible targets. There are publications describing lncRNA, so they could compare the results (Rajan et al, 2020; Tinti et al, 2021; Bento et al, bioRXiv); to mention some). Even a comparison with procyclic data (from CL) would be interesting. With these data, they could expand their bioinformatic analysis.

T. brucei not always italic

Round 2

Reviewer 3 Report

Thanks for the modifications.

The paragraph that goes from lines 165 to 174 should be removed (if I understand correctly it was replaced by the next one).

Reviewer 4 Report

The authors have answered my questions and addressed my concerns.